# Usage of Vertical Fisheye-Images to Quantify Urban Light Pollution on Small Scales and the Impact of LED Conversion

**DOI:** 10.3390/jimaging5110086

**Published:** 2019-11-18

**Authors:** Stefan Wallner

**Affiliations:** 1Department of Astrophysics, University of Vienna, Türkenschanzstrasse 17, 1180 Vienna, Austria; stefan.wallner@univie.ac.at; 2ICA, Slovak Academy of Sciences, Dúbravská cesta 9, 84503 Bratislava, Slovak Republic

**Keywords:** light pollution, digital imaging, lighting quality, LED, colour temperature, all-sky, CCT, fisheye

## Abstract

The aim of this work was to develop an easy and quick technique for characterizing various lighting situations, that is, single lamps or illuminated signs and to quantify impacts on small scales like streets, buildings and near areas. The method uses a DSLR-camera equipped with fisheye-lens and the software Sky Quality Camera, both commonly used as part of night sky imagery in the light pollution community, to obtain information about luminance and correlated colour temperature. As a difference to its usual build-up, observed light emitting sources were captured by pointing the camera towards analysed objects, that is, images were taken via vertical plane imaging with very short exposure times under one second. Results have proven that this technique provides a practical way to quantify the lighting efficacy in a certain place or area, as a quantitative analysis of the direct emission towards the observer and the illumination on surroundings, that is, street surfaces, sidewalks and buildings, was performed. When conducting lamp conversions, the method can be used to characterize the gradient of change and could be a useful tool for municipalities to find the optimal lighting solution. The paper shows examples of different lighting situations like single lamps of different types, also containing various luminaires, illuminated billboards or buildings and impacts of the lighting transition to LEDs in the city of Eisenstadt, Austria. The horizontal fisheye method is interdisciplinary applicable, for example, being suitable for lighting management, to sustainability and energy saving purposes.

## 1. Introduction

For many years, the amount of artificial light at night has increased worldwide and so has light pollution. Observations show that 83% of the world’s population live under a light-polluted sky [1] and that there still is a growth of artificially lit areas on Earth by 2.2% per year [2]. This results in an impact noticeable not only for astronomers by losing visibility with increasing night sky brightness [3,4], but furthermore, artificial light interferes with ecosystems on our planet. Nature and environment are affected negatively as they suffer from skyglow over cities [5,6,7,8]. Even aquatic ecosystems suffer from light pollution as observed [9,10]. However, also impacts on human health cannot be neglected, as sleep deprivation is only one among many issues found [11,12,13]. In past scientific works, one can find suggestions on an environmentally friendly usage of artificial light, while still providing possibilities of energy savings. Consequently, there shall be a focus for global change in the 21st century [14,15,16]. One chance is by changing current existing and mostly outdated lighting situation systems in cities from inefficient lamps of past generations (High pressure sodium, Low pressure sodium, etc.) to Light Emitting Diodes (LEDs). With a higher efficiency, they provide the possibilities of controlling colour temperatures or being able to dim during night-time, to only name a few. However, their use—not only in public but also indoor light and luminous screens of TVs, laptops and smartphones—can become dangerous if using a great deal of light in short wavelengths (blue) which has even more negative effects on human health or the environment [9,13,17]. A lot of cities worldwide changed their street lighting to LEDs already and the impacts of those changes were observed [18,19] or modelled [20].

The most commonly used measurement technique for quantifying light pollution are ‘Sky Quality Meters’ (SQMs) [21,22], giving the opportunity of one-dimensional photometric measurements by detecting the brightness of the sky’s zenith luminosity. Sometimes, if various devices are available, they are grouped together to form light monitoring networks [23,24]. Other methods involve all-sky images, which provide two-dimensional images of the night sky [25,26,27,28,29,30], satellite data, especially to quantify the development of light pollution on global scales [1,2,4,31] or theoretical simulations to calculate the behaviour of light in our atmosphere [32,33,34]. All methods do have pros and cons [35]. In general, impacts of entire cities or districts were analysed to see, for example, a gradient in skyglow.

The aim of this work is to find an easy and quick method by using a currently used technique in light pollution communities, the all-sky method, for quality analysis on small scales. This includes either single lamps of different types or the impact of artificial light on objects as houses, buildings or streets. This should help characterizing lighting situations easily on a low time scale.

## 2. Materials and Methods

### 2.1. Location

To observe on the one hand various examples of lighting situations and lamps and on the other hand the impact of a LED conversion, the city of Eisenstadt, Austria (N 47°50′46.93′′, E 16°31′40.66′′) was chosen. The capital city of the Austrian municipality Burgenland is home to total of 14,000 inhabitants. In 2018, the local government decided to undergo a transition of the existing public street lighting system, from mostly High Pressure Sodium Lamps (HPS) to more modern and energy saving LEDs. The conversion started in mid 2018 and was finished in April 2019. More than 2400 lamps were changed to LEDs, only few HPS lamps were maintained in the inner city for historical reasons. As chosen by the municipal authorities, the colour temperature for all LEDs is 3000 K, being also fully shielded with ULOR = 0 (Uplight Output Ratio), that is, no light emitted to the upper hemisphere.

With having changed most of the public street lighting, there still exists a variety of lamps distributed over the city. Especially when observing areas not controlled by the local authorities, for example, schools or parking spaces, lamp types can differ significantly from those changed. Some of them will be replaced in the future, for some there are no plans of being exchanged so far. The great number of lighting situations, especially light sources not affected by the LED conversion, makes the city of Eisenstadt an optimal example of capturing various points of reference.

### 2.2. Instruments & Images

Images were taken by a digital single-lens reflex camera (Canon EOS 5D Mark II) with fisheye-lens. This method is more commonly used for obtaining all-sky images of the night sky to analyse its brightness, the impact of cloudy sky conditions or the identification of light domes close to the horizon from light emitting sources [25,26,27,28,29,30]. Because neither of these are important parameters for the purpose of this work, the camera was not pointed onto the sky but rather in vertical directions. This approach was used before [28], here it is used inside of a city observing not traces on the sky but to quantify impacts of single lamps, illuminances of buildings, advertisements and impacts on streets. Hence, this method is used on a far smaller scale being able to characterize single light pollution sources and impacts on surroundings.

Various types of images were taken: (i) images of six different lamp types, (ii) images of two illuminated buildings, (iii) images of two luminous advertisements, (iv) images of a street before and after the LED conversion. As some of the images were captured before and some after the transition of public street lighting, they were obtained between July 2018 and October 2019. The image captions each include the exact recording dates. The ISO of 1600 and focal ratio of f/3.5 remained constant for all images. However, because of the very bright light emitting sources, exposure times can be kept very short, under one second per image. Therefore, a tripod is not absolutely necessary. All images were taken under clear sky conditions. Lamps and other emitting sources were always centred as best as possible.

Since further analysis steps shall also provide correlated colour temperatures (CCT) of each lighting situation, also spectra of lamps—see Section 3.1—were obtained, creating the possibility of comparing software results to spectra measured. In this regard, the spectrometer Mavospec Base by Gossen (Nuremberg, Germany) was used. However, this instrument is only used to allow a comparison and is in principal not part of the measurement technique described in this work.

### 2.3. Software & Analysis Method

To analyse all images in detail, there is the need of a software which is able to give luminance and CCT data as output. One which is capable of doing so is “Sky Quality Camera” (SQC) developed by Andrej Mohar from Euromix d.o.o. (Ljubljana, Slovenia). The primary goals of this software are to process all-sky images of the night sky to quantify the impact of artificial light at night and to identify certain light emitting sources close to the horizon. The software is therefore a common tool in light pollution communities and used continuously when analysing all-sky images. Furthermore, like mentioned above, SQC was also used for horizontal fisheye-images before, but as a difference, not in an urban environment or quantitatively [25,26,27,28,29,30].

Important settings of the software being used for this work are—no subtraction of brightest stars or Milky Way (since no sky is observed), altitude of sky border is 0 degrees (to allow the full field of view being analysed), clear sky (no automatic detection of possible clouds).

At this point is must be mentioned that exact data values, like mcd/m^2^ for luminance or Kelvin for CCT, can be obtained interactively by the user, meaning that when moving the mouse over the resulting images, values for certain positions, for example, the lamp’s head, are displayed. If no further explanation is included, this is how some values, also possibly beyond the illustrated scales, were read.

The approach of this work, to use luminance and CCT analyses for single light emitting sources shall be a new method to obtain information not only about the sources itself but to compare different systems for a possible conversion, especially LEDs. Analysing the gradient of luminance and CCT temperature will also be observed by subtraction of SQC images [29]. This shows the development of these parameters in an optimal way. The subtraction is a tool provided by the software, images must be aligned to each other before a subtraction is performed. Alignment can be performed in the software via manual rotation in three axes.

As exposure times are very short, signal to noise ratio rapidly decreases for dark areas. This means in effect, that this technique cannot be used to quantify impacts on the night sky. Although, a certain glare caused by the light emitting sources possibly pointing towards the sky can be observed.

The used camera is calibrated to the software.

## 3. Results of Various Lighting Systems

### 3.1. Lamps

Hereafter, six examples of different lamp types are displayed and analysed. Every lamp was also photographed during daytime to receive an impression of its shape. Furthermore, the spectrum was collected like described in Section 2.2. The fisheye image which is analysed by SQC software is also shown with the resulting outcomes.

#### 3.1.1. HPS Lamp

The High Pressure Sodium (HPS) lamp, like captured in Figure 1a, was the method of choice for public street lighting for many years. Not only Eisenstadt but for many cities around the globe it used to be or still is the most ubiquitous type. In the near past, with the development of LEDs, doubts raised on it, especially its low luminous efficacy. The spectrum of the lamp, visible in Figure 1b, shows the typical peak of the sodium D-line at 589 nm, the visible result is a golden-orange light. The colour temperature measured was 2037 K. The lighting situation captured, seen in Figure 2a shows a typical street lighting situation in Eisenstadt before its change. The image was taken midway between two HPS lamps.

Luminance data of the resulting image, provided by the SQC software, are shown in Figure 2b. Values at the HPS lamp’s head reach numbers up to 7250 mcd/m^2^, on the street between 100 (at observer) and 2200 mcd/m^2^ (under lamp). It occurs, when looking at nearby standing trees, that the lamp, as typical for HPS lamps, does not show full cut-off behaviour by far. There is high amount of emission to the sides, also visible at the building in the background. A comparison to another HPS lights inside different shaped lamps is described later in Section 3.1.3 and Section 3.1.5. The resulting image also shows structures on the night sky, looking like ice pillars. As this was definitely not possible as temperatures were way to high, it seems this is an effect of saturation and low signal to noise on the night sky. However, as we do not want to analyse the night sky, this error can be neglected.

Figure 2c shows the correlated colour temperature matrix for the image. On the illuminated street, CCT analysis displays values between 1800–2200 K which matches the measurement of the lamp. For the lamp’s light emitting head SQC shows higher values up to 2300 K. This can be explained by a saturation happening by its high luminosity. It can be concluded that the CCT analysis can only be applied to illuminated surfaces and glare around the light, as latter can be seen in the luminance data and still carries information about the colour temperature as seen in the CCT data.

#### 3.1.2. LED Lamp

Since mid 2019, more than 2400 Light-Emitting Diodes replace HPS lamps for the public street lighting in Eisenstadt. LEDs do carry a lot of advantages, being cheaper in the maintenance, having the highest luminous efficacy on the market (up to 350 lm/W as theoretical maximum), for the first time being able to dim single lamps and generate colour temperatures as wished. This in its turn means, that LEDs can also be reason of danger for the environment as, for example, over-illumination is produced more easily. Furthermore, as more light is emitted at lower wavelengths, that is, blue light, has negative effects on animals, nature and humans as discussed above. Figure 3a shows the type of head now distributed over the whole city. The spectrum, shown in Figure 3b, definitely matches LED-like spectra with a colour temperature of 3201 K measured. The image taken for analysis is shown in Figure 4a.

All LEDs installed are full cut-off, meaning that there is no emission to the upper hemisphere. This is underlined by the luminance matrix in Figure 4b. Higher luminance values only appear on the street, the sidewalk and a small area surrounding the lamp. The building standing right to the lamp is only illuminated very weakly. In the image taken, Figure 4a, it is only slightly visible. The building displays luminance values up to a maximum of 70 mcd/m^2^, the sidewalk between 90–600 mcd/m^2^ (observer to below lamp) and the lamp’s head up to 2256 mcd/m^2^. Furthermore, a tree, which can be seen at the upper right edge of the image, appears only to be slightly illuminated as an impact of reflection on the street. The ratio of illuminance of the street between lamp and observer seems to be more uniform compared to the HPS lamp.

The CCT data, illustrated in Figure 4c, show great distribution. The street and sidewalk show values between 2600–3000 K, the green area (grass) on the right to the lamp between 2700–3200 K, the building between 2800–3300 K. The red stroke in the centre of the image shows a person wearing a red uniform. It seems that for the LED lamp, due to its lower luminance values, a mixing in colour temperatures is happening between the light and illuminated surfaces, that is, that effects caused by the wavelength-dependent surface albedo cause the colour temperature of the illuminated ground to be lower than the lamp’s light itself. Subsequently, the CCT analysis is able to display, how strongly this mixing is happening depending on the strength of illuminance. With increasing luminance of lamps, see, for example, the HPS-lamp in Figure 2c, also the mixing effects decreases as surface albedo effects play less role. When comparing a LED with higher luminance (see later in Section 3.1.4) the colour of the lamp is dominating also on the ground.

#### 3.1.3. Ball Light

In the city, this type of lamp, as pictured in Figure 5a, was frequently used for illumination around buildings, parking areas and pedestrian zones. Today only a few of them are left. With a colour temperature of 1856 K, as measured by the spectrometer, its spectrum, illustrated in Figure 5b, shows a dominant peak in the orange wavelengths and can be described as HPS-like, comparing to Figure 1b. The slight difference to the HPS spectrum seen before, can be explained by a frosted glass film around the ball light. The biggest disadvantage of a ball light is the emission of light to all angles and therefore wasting energy especially by emitting to the upper hemisphere as can be seen in Figure 6a. As the mounting is affixed at the bottom it can be assumed that there is even more light emitted upwards than downwards. An advantage of this lamp is the colour temperature, which only includes a neglectable peak at lower wavelengths at 500 nm.

Results of performed analysis, be it the luminance or the CCT data, underline the features of the ball light discussed before. Figure 6b shows that high luminance values are not only found on the footpath or parking area but also up to the building’s top. This confirms the fact of light emission to all angles. Values at the lamp’s head rise to around 3500 mcd/m^2^, whereas the footpath under the lamp is lower illuminated (<540 mcd/m^2^) than the building right behind the lamp (<750 mcd/m^2^). Also, it is of great concern that the tree, standing left to the lamp, also is illuminated very strongly, showing values up to 200 mcd/m^2^.

The correlated colour temperature, displayed in Figure 6c, differs slightly from the spectrometer’s value. SQC shows a CCT of around 2000 K for all illuminated surfaces. When displaying averaged CCT values in areas depending on the angles from the image centre, between 0° and 30° the result is 2049 K, whereas between 0° and 90° (whole image) shows nearly no difference with 2031 K. Consequently, this is again strengthening the nearly uniform emission to all angles. Results, especially the colour temperature, give evidence that the ball light contains a HPS lamp.

#### 3.1.4. Tube Lamp

At first view, one might be led to believe the tube lamp, captured in Figure 7a, seems to emit light like a fluorescent tube. However, Figure 8a shows that the emission only occurs at the top of the pipe shaped light. This type is not widely spread and only a few are used for a car park. Its spectrum, illustrated in Figure 7b, confirms that this lamp contains a LED with a colour temperature of 2921 K. Comparing its appearance to the new LED street lighting, for example, displayed in Figure 4a, this becomes even clearer. Although, the emission of light only happens at the top of the lamp, it raises the question, if the type of construction of this lamp is necessary. There is the possibility that it was originally designed for fluorescent tubes, which would increase the apex angle of emission compared to a LED located only at the top extremely.

Luminance data, see Figure 8b, illustrate that emission of light is not only pointed to the lower hemisphere as the tree standing behind the lamp is fully illuminated. Values at the lamp’s top appear to be higher, up to 4200 mcd/m^2^, than the ball light in Figure 8a, the area close to the lamp (~2 m around it) between 1000–1500 mcd/m^2^.

Other light emitting sources can be observed in the background, like a billboard left of the lamp. This is clearly visible when examining the CCT analysis in Figure 8c, as there are different colour temperatures. Compared to the HPS lamp, also here SQC displays a colour temperature of ~2300 K for the lamp itself, which is far from its real value. Its glare around the lamp’s head, which can be seen in the luminance data, gives more realistic values between 2850–3100 K. In conclusion, the CCT analysis provides the possibility of a separation of different light sources, also displaying the impact of each. This effect is not possible when only observing the luminance matrix.

#### 3.1.5. Lantern

Around 70 lanterns, like pictured in Figure 9a, were not affected by the conversion to LEDs in Eisenstadt. They are located in the inner city, illuminating the pedestrian zone and will be maintained for historical reasons. Its spectrum, illustrated in Figure 9b, appears to be the same as obtained from the ball light in Figure 5b. Also, the measured colour temperature, 1896 K, underlines this. Consequently, the lanterns also include HPS lamps. Figure 10a shows the lighting situation at the pedestrian area, when looking to the left of the centred lamp, a great number of other lanterns are visible.

As expected, based on the lamp’s shape, the luminance data, displayed in Figure 10b, show that also the lantern enlightens in nearly all angles as buildings in the near surrounding appear to be strongly illuminated. Showing values around 13,500 mcd/m^2^, the lantern is more luminous than the ball light. This also has an impact on the wall behind the lamp, having luminance values up to 3500 mcd/m^2^. In comparison to the HPS lamp used on the street, the luminance data show a more uniform distribution of the illumination to all sides, whereas in Figure 2b some objects on the sides of the street, like a tree on the left side of the image, only display lower luminance values. Consequently, the light emission of the HPS lamp measured is more orientated to lower angles than observed here.

The CCT analysis in Figure 10c illustrates the same pattern as the engulfing glare of the ball light in Figure 6c. CCT values of the illuminated surfaces are distributed between 1800–2200 K.

#### 3.1.6. Ring-Shaped Lamp

The ring-shaped lamp, captured in Figure 11a, appears to emit no light to the upper hemisphere due to its shape. The measured spectrum, visible in Figure 11b, matches a fluorescent lamp with an obtained colour temperature of 3660 K. There are only few of those installed in the city of Eisenstadt, the one observed here is part of a parking area, as seen in Figure 12a.

Figure 12b shows the luminance matrix of the analysed image and gives indications about the cut-off behaviour of this lamp type, especially when looking at the tree standing right next to the lamp. Here it must be mentioned, that it is illuminated by a second ring-shaped lamp, which is not visible in this image. Still, a clearly observable gradient in luminance values is displayed, as the bottom of the tree shows values up to 290 mcd/m^2^ and the top up to 28 mcd/m^2^. Consequently, there is a factor of 10 in luminance between the bottom and the top of the tree. The lamp’s head displays up to 4400 mcd/m^2^.

Also, CCT data, as seen in Figure 12c, can be used to discuss this behaviour. The colour temperature of the lamps glare shows values between 3400–3800 K. Through mixing with colour temperatures of surfaces, the parking area shows values around 3000–3400 K, 3500–3600 K where white lines mark the parking spaces, 3600–3900 K at the grassed area. Green leaves of the tree show values up to 4500 K. Here, also the tree must be discussed in more detail as values of the lamps colour temperature are only visible on the tree up to the height of the lamp, the upper part of the tree shows in general much higher values.

Comparing to the full cut-off LED lamp in Section 3.1.2, results of luminance and CCT matrices show that emission of light is also concentrated to the lower hemisphere, whereas here emission appear to be 90° as values on the tree standing near the lamp(s) strongly decrease above the lamp’s height. By definition of the Illumination Engineering Society (IES), full cut-off LED are predestined to emit no light at or above horizontal, that is, 90° above nadir and limited to <10% intensity at or above 80°.

### 3.2. Buildings

In the following subsection, there are observations of bigger scaled impacts of grouped lamps, still small compared to whole cities. Two examples of illuminated buildings were captured, first a Market Hall and then Castle Esterházy, a touristic magnet of Eisenstadt. In the former case, the Market Hall is not illuminated every night, that is, there was no continuous periodicity found, whereas the Castle is illuminated throughout a year. Missing local laws regulating the brightness of building facades, there is no continuity in illuminated surfaces in the city. However, the second example, Castle Esterhazy, appears to be the brightest illuminated building in Eisenstadt.

#### 3.2.1. Lower Luminance—Market Hall

Figure 13a shows the lighting situation in front of the Market Hall. The much stronger illuminated Castle Esterházy is located in the near surrounding but has no dominant impact on the imaged Market Hall. To prove this, also the castle was captured on the right edge of the image. Illumination of the Market Hall is executed by lamps installed on the ground emitting light upwards towards the wall. The wall is painted with bright yellow and orange colours, which is going to show impacts on the analyses as shown later.

Luminance data, as illustrated in Figure 13b, show great distributions as some parameters must be considered. At first, it becomes clear that the observation angle plays an important role. When looking at the pillars, which are directly illuminated by lamps from the ground, luminance values increase with increasing distance to the image centre, which has a passage as significant feature. The two pillars around this passage both show values between 3500–3650 mcd/m^2^, the second pillar left of the passage 3850–4000 mcd/m^2^, right 3700–3800 mcd/m^2^, the third left 4250–4450 mcd/m^2^, right 4000–4100 mcd/m^2^, to give only few examples. The seventh pillar left of the passage, which marks the last one, even shows values between 6200–6350 mcd/m^2^. This effect can be easily explained as bigger observation angles are overlying impacts of lighting situations with more than one light emitting source. Subsequently, for the analysis of buildings and streets with different distributed light sources standing near each other, this must be considered. Below, in Section 4, the impact on a street will be observed but as a subtraction is being chosen for analysis, the whole image can be used to find out valuable and reliable information as subtraction is also performed at overlying illuminances at the edges. The ground, approximately one meter ahead of the wall, is nearly evenly illuminated with luminance values between 250–300 mcd/m^2^, values in front of the observer (approx. 5 m from the Market Hall) still display around 70 mcd/m^2^. This shows huge impact of lamps even if emitting light only upwards due to reflection and over-illumination.

CCT data, as displayed in Figure 13c, show a great variety of colour temperatures. Like mentioned before, one reason is the yellowish painted wall which mixes with the colour of the light. It appears that strong illuminated surfaces, like above lamps, show values of 2300 K, not strong illuminated surfaces, where the wall’s colour is visible, between 2600–2650 K. Also, values >3000 K appear at windows and doors which are not illuminated, the passage even displays values >5500 K as a blue billboard is located inside.

#### 3.2.2. Higher Luminance—Castle Esterházy

Castle Esterházy marks the landmark of Eisenstadt and therefore appears to be the most touristic visited place in the city. Its outer façade is illuminated every night, throughout a year, playing a big role on the light dome existing above the city. Since its illumination is very strong, Figure 14a shows the image taken at an exposure time of only 1/30 s. The size of this manifest shows one of the great advantages of fisheye lenses, having no problems to capture the whole building while standing very near. The image here was taken at ~15 m distance to the main entrance. The sources of its illumination are various—there are lamps installed on the outside of the balcony enlightening the facade upwards, there are lamps around the main entrance illuminating from the ceiling downwards, there is an illuminated billboard above the main entrance, there are two headlamps located in front of the castle enlightening the left and right parts of the castle and there are two very strong headlamps located on the neighbouring building illuminating the two towers from a distance of approximately 50 m. In conclusion, there is a great number of sources, causing the castle to be strongly illuminated.

By looking at the luminance data of Figure 14b some can suggest nearly continuous luminance values over the building. Detailed analysis shows the opposite, as the scale provided by the software, is now by far too low. This is underlined when displaying the following: the right edge of the castle shows values 1000–2500 mcd/m^2^, the left part 3000–20,000 mcd/m^2^, the centre part 6000–48,000 mcd/m^2^, the right part 5000–8000 mcd/m^2^, the right edge 1000–4000 mcd/m^2^, the right tower 4800–5800 mcd/m^2^ and the advertisement above the main entrance <44,000 mcd/m^2^. The results show that no uniformity in illumination is existing and, for example, the left part of the castle is stronger enlightened than the right part. All values were analysed on the building’s surfaces, not on windows, doors or passages. The left tower was restructured as a scaffold is visible. Therefore, the left tower cannot be analysed. As light emitting sources, that is, the lamps, are clearly separated from each other, there is no overlying in illuminations like seen in Section 3.2.1. The data also show great impacts on the ground in the surrounding of the castle, having 350–500 mcd/m^2^ on the grassed areas in front or between 2000 mcd/m^2^ on the footpath in front of the main entrance to around 320 mcd/m^2^ at the observer.

As envisaged, the CCT data, displayed in Figure 14c, also show that the lamps used for illumination of the castle also differ in types and colour temperatures. Again, it must be considered, that mixtures in colour temperatures through the yellowish coloured wall are resulting. The centre part of the castle shows nearly a uniform distribution of values between 3000–3200 K, the left part between 2500–3000 K, the right part 2700–3300 K and the right tower 3100–3600 K. Comparing the different parts of the building again shows that no uniformity in illumination is existing.

### 3.3. Billboards and Signs

This section describes analyses of illuminated signs in the city. The first one is an illuminated logo lettering, the second one a LED video wall. Both are enlightened throughout a year in Eisenstadt.

#### 3.3.1. Monochromatic Sign

As Figure 15a displays, the sign is a monochromatic logo lettering of a local market. It is installed beside one of the most frequented roads of the city. Its observed spectrum, shown in Figure 15b, underlines its monochromatic behaviour. The lettering captured here has a reddish colour, whereas the market also owns another lettering greenish coloured. Latter one is located around the corner on another side of the market but impacts are still visible on the left side of the image.

The luminance matrix obtained by SQC software, visible in Figure 15b, shows values of the sign between 13,500–15,500 mcd/m^2^. The ground in front of the sign seems to be only slightly illuminated, displaying <20 mcd/m^2^, reflections higher up, for example, at the second left wooden cottage in front of the sign <170 mcd/m^2^. The ground in front of the cottages almost is not illuminated due to being shadowed by them. The tree on the right edge of the image shows an interesting result, as the side looking towards the sign is lower enlightened than the side pointing towards the street and being illuminated by a LED lamp there.

CCT data underline latter discussed result, as, like seen in Figure 15c, the tree also shows a different colour temperature on the side looking to the street (~2400 K) than to the sign (1500 K). The illuminated letters show a uniform colour temperature of 2300K whereas the glare of 1500 K. As latter displays the found 1500 K also very uniformly, it seems that the lower end of scale still seems to be too high as values could also distribute below. CCT data from the ground show that even if there is nearly no luminance at all, as mentioned before due to shadow of the wooden cottages, there are still effects of the illuminated sign as values are around 1500 K again.

#### 3.3.2. LED Video Wall

As a second example of enlightened street advertisements, a LED video wall was captured. In total, three images of different displays were taken to also see the impact of different visual displays with different colour temperatures. Figure 16a–c show luminance matrices of the different advertisements 1–3, Figure 16d–f the related CCT analyses of advertisements 1–3.

Results clearly show different impacts on the surrounding if watching different displays. Advertisement 1, Figure 16a,d,g, seems to have the biggest impact as its glare leaves marks in front of the night sky. When observing the area of 0° to 13° around the centre of the image (where billboard is located), luminance values spread between 7000–14,000 mcd/m^2^ and CCT values between 2200–5400 K. These big differences are not surprising as they are strongly dependent from colour of background and used letters on the display. Advertisement 2, Figure 16b,e,h, shows luminance values in the mentioned area between 5000–13,500 mcd/m^2^ and colour temperatures between 2400–7600 K, advertisement 3, Figure 16c,f,i, 1500–11,000 mcd/m^2^ and 1900–7000 K.

## 4. Results of LED Lighting Transition Analysis

To analyse impacts of the LED conversion performed in Eisenstadt, ‘overview’ images before and after the transition were captured from a high-rise building. The images show a street and its lighting situation being able to see impacts also on nearby buildings, plants and sidewalks. Later the images get subtracted, a tool which is implemented in the SQC software, to show changes in more detail.

### 4.1. Observations before LED Conversion

The image taken before LED transition started in Eisenstadt, was obtained in July 2018. It is displayed in Figure 17a. Here one must say, that moon was ~12° above the horizon, being 91.6% illuminated. Meteorological conditions were the reason why later images, after that lunar cycle, were not possible, as conversion started directly after. Consequently, the moon will play a role on the night sky in both luminance and CCT data but not on the lighting situation of the street or its analysed surrounding, as the high-rise building was blocking any moonlight. The moon was located behind the building (viewing direction to the west) and only sparsely above the horizon.

Figure 17b shows the luminance matrix and on the left edge there are higher luminance values on the sky visible, which are now a result of the moon. However, also light domes from light emitting sources farer away are apparent. Still, coming back to the lighting situation of the street, it becomes clear that, unsurprisingly, the highest luminance values, beside the lamps themselves, appear to be directly below the lamps. Two lamps are located in front of the observer, the street’s surface below both showing values between 1000–1200 mcd/m^2^, the surface midway between 80–160 mcd/m^2^. Furthermore, looking at the lamp to the right, there is also an impact on the thuja plants below the lamp’s head, being illuminated up to 2000 mcd/m^2^. Despite being ‘protected’ by those thuja plants, the building located in the centre image, shows great impact too, being illuminated up to the top. The highest luminance values displayed on the house’s wall are 45 mcd/m^2^. Compared to Section 3.1.1, this situation again shows the emission of the HPS lamp to also angles of the upper hemisphere.

Unsurprisingly, CCT data, illustrated in Figure 17c, show the street being dominated by the HPS lamp having nearly uniform values between 2000–2100 K. The house also shows values between 1900–2300 K, underlining the impact of light coming from the street. Looking at thuja between the lamps, there is a great distribution of CCT data, which is a result of low illumination. Still, also the thuja near the lamp to the right are dominated by values around 2000–2400 K.

### 4.2. Observations after LED Conversion

Figure 18a shows the pictured lighting situation after the LED conversion was finished in July 2019. Being taken with the same exposure times, one can clearly see major differences, as, for example, the building in the image centre appears to be lower illuminated and impacts on the night sky, that is, light domes from various city parts, seems to have darkened.

Luminance data, illustrated in Figure 18b, underline these impacts. Values under the lamps now display 350–600 mcd/m^2^, midway of the lamps 30–130 mcd/m^2^. Still, there is also an impact on the thuja, caused by the lamp to the right, of <1300 mcd/m^2^. Also, the building again appears to be illuminated but much lower than seen before, with values around 18 mcd/m^2^ at the top and 2–5 mcd/m^2^ at the bottom.

The resulting CCT data, shown in Figure 18c, displays the street being engulfed in a new colour with values between 2700–3100 K. The house now shows great distribution in CCT values and it is not possible to see a threshold of values on its surface. However, at the top it seems that values spread between 2400–3400 K, whereas at the bottom at much higher ones.

### 4.3. Comparison and Subtraction

Figure 19 shows an optimal way of comparison between the lighting situations before and after the LED conversion. Subtracting the image before the transition from the image after transition shows how values changed in respect to the current condition.

In Figure 19a,b, one can see the change in luminance data in two different scales. In first, with lower scale, it is clearly visible that at most surface areas luminance has decreased (green coloured), only few, especially the sidewalks, have increased. Decreasing values on the left of the night sky again show impact of the moon, being on the sky at the image captured before the transition.

To quantify areas with higher impacts in more detail, also a higher scale was chosen. Results show that on the street, luminance decreased stronger below lamps, on the wall of the building in the image centre, the thuja and even at other city parts visible on the right side of the image. Increased luminance can be observed at sidewalks and small street parts. Both can be explained by the full cut-off behaviour of the new installed LED lamps. Luminance values under lamps on the street show a decrease of up to 770 mcd/m^2^, midway between the lamps up to 50 mcd/m^2^. The house in the centre of the image shows a decrease of 35 mcd/m^2^ on the top and between 10–15 mcd/m^2^ in the centre. The sidewalk is analysed by an increase between 40 to 120 mcd/m^2^.

CCT data, shown in Figure 19c, confirm the expectations as colour temperature appears to be increasing with the transition. Comparing to Section 3.1.1 and Section 3.1.2, changes in CCT must be around 1000 K, which is underlined in the SQC result. A stronger increase is shown at areas which are lower illuminated after the change in lighting system.

Table 1 shows resulting values of the subtraction for luminance and CCT at different areas depending on the angle from the image centre. As both lamps located in front of the observer are located at around ~60° from the image centre, also the area 59°–61° was included.

## 5. Discussion

The results obtained showed that vertical fisheye imaging and analyses performed via Sky Quality Camera software can provide lot of valuable information about light emitting sources and their impacts on surroundings. Especially the wide-angle characterization enabled the possibility of examining various luminaires in different lighting settings. Moreover, due to the wide field optics, there was also the opportunity of analysing facades, particularly of buildings greater than the field of view if using a ‘normal,’ that is, non-fisheye, camera lens while standing near to keep the signal high. SQC is capable of obtaining luminance or CCT values interactively, allowing local studies with respect to uniformity of illuminance or identifying extreme values, easily, as also studied in the past [36]. Furthermore, the technique is able to perform quantitative analyses of not only the emission of light sources but especially their direct emission towards the observer. The analysis of a video wall has shown that the impact on its surrounding depends on the display content and can vary strongly by one and the same LED billboard. Especially luminance data illustrate that it is the dominant colour of the advertisement’s background which specifies the strength of illuminance on the sidewalk, street and so forth. Therefore, if needed to characterize such video walls, it is necessary observing the ad showing measurably the strongest luminance values. After the transition from mostly HPS lamps to LEDs in Eisenstadt, it was possible to analyse the impacts of this change at one location. Subtracting images taken before and after the conversion provides a great possibility of comparison and to quantify impacts of old versus new light sources on surroundings. As expected, higher colour temperatures, an increase of around 800 K from HPS lamp to LEDs and mostly lower luminance values, a decrease of up to 770 mcd/m^2^, were the result of the analysis.

There are limitations to the analysis as high luminance values, that is, values mostly above 4000 mcd/m^2^, can be followed by a saturation of the image, visible in Figure 8 and Figure 10. Especially heads of lamps seem to show different CCT values than their glare or illuminated surfaces, which are more realistic compared to spectrometer measurements. Consequently, this must be considered if trying to find the colour temperature of a lamp. If obtaining colour temperatures of illuminated surfaces, one must take into account that a mixing of colour temperatures of either the emitted light or the painted surfaces occurs. Subsequently, the CCT of an illuminated surface is rarely in exact accordance with the light coming from the lamp. As SQC is generally designed for imaging the night sky, the luminance scale’s thresholds applied to the images can be too low, especially when capturing illuminated signs or buildings. Hence, differentiating of luminance values above 4000 mcd/m^2^ is not possible by only looking at the resulting matrices. It may well be the case if using the interactive software as, when moving the mouse over the image, values above this threshold are still displayed.

The paper shows results of only few usages of vertical fisheye images as they can be used in more multifaceted ways for further study, for example, lamp dimming in different intensities or the impact of different heights of lamp heads. These are only two examples of far more ideas, how the technique can be used in future works. It is interdisciplinary applicable, as it can be used to quantify impacts of artificial light at night, especially single light emitting sources for studies in ecology, sustainability, lighting management and far more. All images analysed were taken under clear sky conditions to avoid external influences and being able to concentrate on impacts on the ground. Still, it is of great interest for future works to investigate the method described in this work also under different meteorological conditions. It could be expected that, for example, foggy conditions are followed by major impacts on luminance data, whereas an overcast sky on CCT data, as generally seen on night sky imagery [26].

## 6. Conclusions

There were various examples of light emitting sources, illuminated objects and lighting situations being analysed with the Sky Quality Camera software via vertical plane imaging. Luminance matrices give the possibility of receiving an impression on the luminosity of light emitting sources and their illuminated surroundings. CCT analyses can not only distinguish but also quantify impacts of different light emitting sources, also when being grouped together. Examples of three different shaped lamps, all containing HPS bulbs, were identified having various impacts on nearby buildings, streets and so forth, depending only on their emission angle. Both luminance and CCT data are able to illustrate the uniformity of illumination of an area around lamps, for example, streets, sidewalks or car parks or on buildings. Furthermore, both can characterize if new installed LEDs or other lamp types are full cut-off or still emitting light to parts of the upper hemisphere.

The technique of using a fisheye lens appears to have both pros and cons. On the one hand, it is easily possible to capture very big buildings on one image without being far away from them and possibly losing signal. Furthermore, its build-up is a commonly used method in the light pollution community, generally imaging the night sky on all-sky images. Therefore, if characterizing single sources, this build-up must not be changed and can be used with much smaller exposure times and even without a tripod. A disadvantage at light emitting sources with a great number of lights and short distance to each other, there is an increasing overlying in illuminances when moving away from the image centre.

In conclusion, the results show that the method of vertical plane imaging and analyses performed via the SQC software is an easy and quick way to obtain information about lighting situations on small scales like the characterization of single lamps or illuminated signs and impacts on streets, buildings and other surroundings. With this method, it is possible to investigate illuminated places, not exclusively urban environments and collect information about possibilities of improvements of lamps installed.

## Figures and Tables

**Figure 1 jimaging-05-00086-f001:**
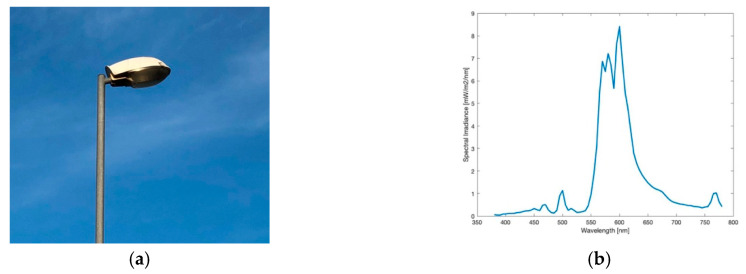
(**a**) Image of the High Pressure Sodium (HPS) lamp during daytime; (**b**) Spectrum of lamp observed.

**Figure 2 jimaging-05-00086-f002:**
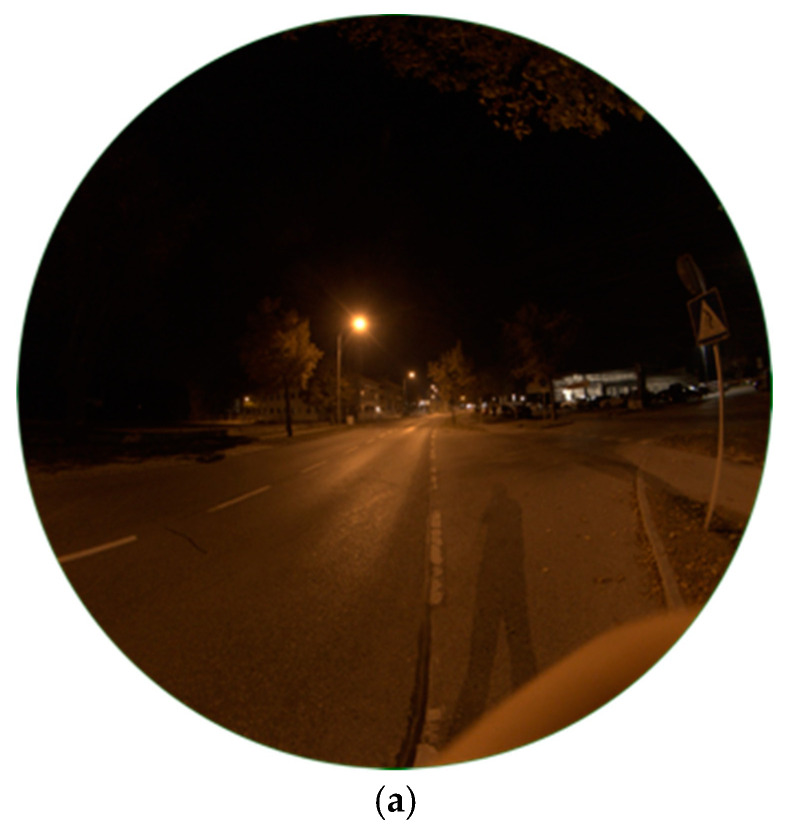
(**a**) Image of HPS lamp during night-time taken with fisheye lens for detailed analysis (date: 10.10.2018, 23:54 h; exposure time: 0.2 s); (**b**) Luminance matrix of image (a) as displayed by Sky Quality Camera (SQC); (**c**) Colour correlated temperature (CCT) matrix of image (a) as displayed by SQC.

**Figure 3 jimaging-05-00086-f003:**
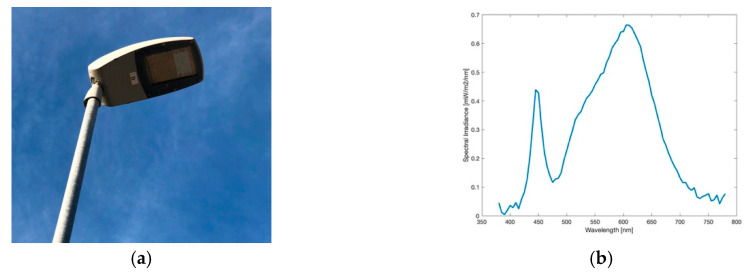
(**a**) Image of the light emitting diode (LED) lamp during daytime; (**b**) Spectrum of lamp observed.

**Figure 4 jimaging-05-00086-f004:**
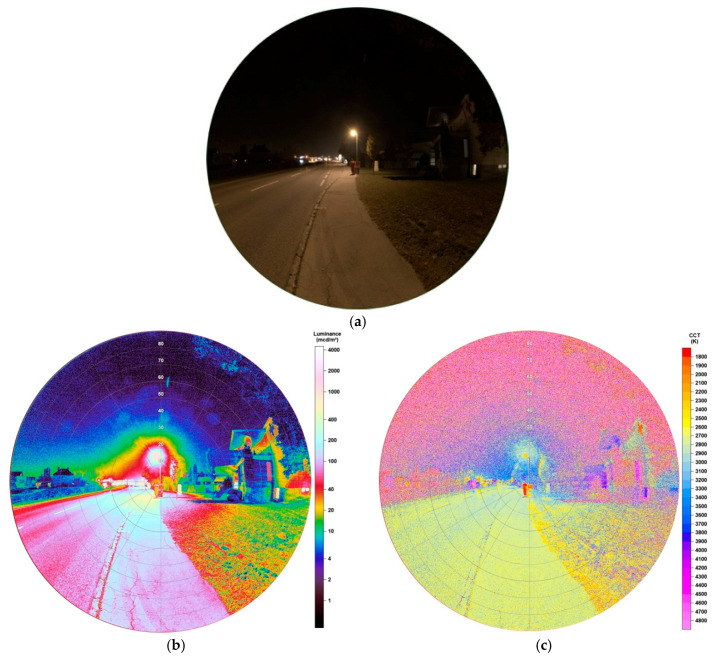
(**a**) Image of LED lamp during night-time taken with fisheye lens for detailed analysis (date: 10.10.2018, 22:57 h; exposure time: 0.6 s); (**b**) Luminance matrix of image (a) as displayed by SQC; (**c**) CCT matrix of image (a) as displayed by SQC.

**Figure 5 jimaging-05-00086-f005:**
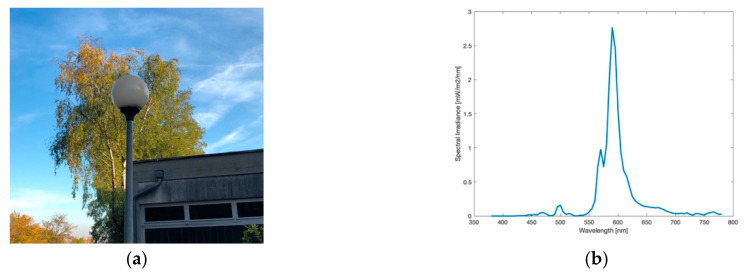
(**a**) Image of the ball light during daytime; (**b**) Spectrum of lamp observed.

**Figure 6 jimaging-05-00086-f006:**
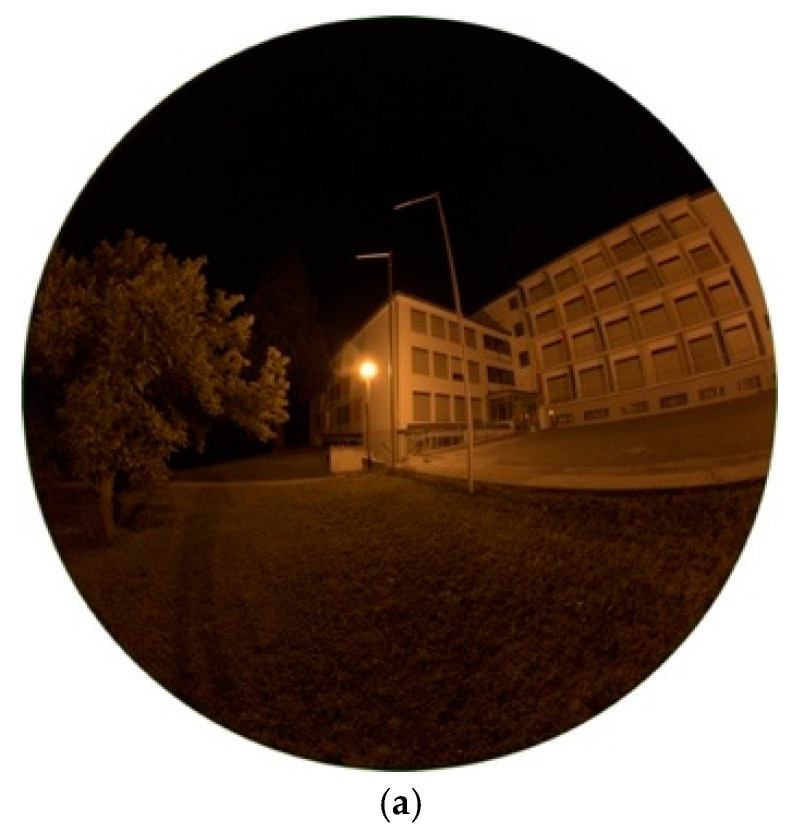
(**a**) Image of ball light during night-time taken with fisheye lens for detailed analysis (date: 29.07.2019, 23:15 h; exposure time: 0.4 s); (**b**) Luminance matrix of image (a) as displayed by SQC; (**c**) CCT matrix of image (a) as displayed by SQC.

**Figure 7 jimaging-05-00086-f007:**
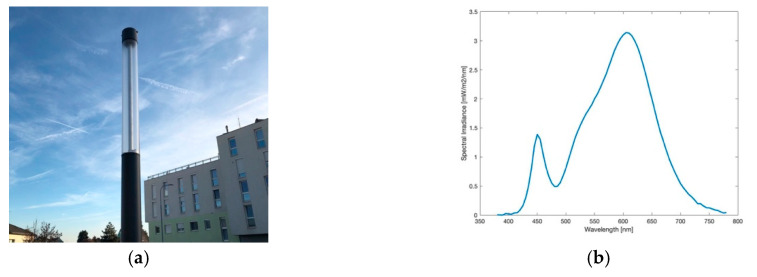
(**a**) Image of the tube lamp during daytime; (**b**) Spectrum of lamp observed.

**Figure 8 jimaging-05-00086-f008:**
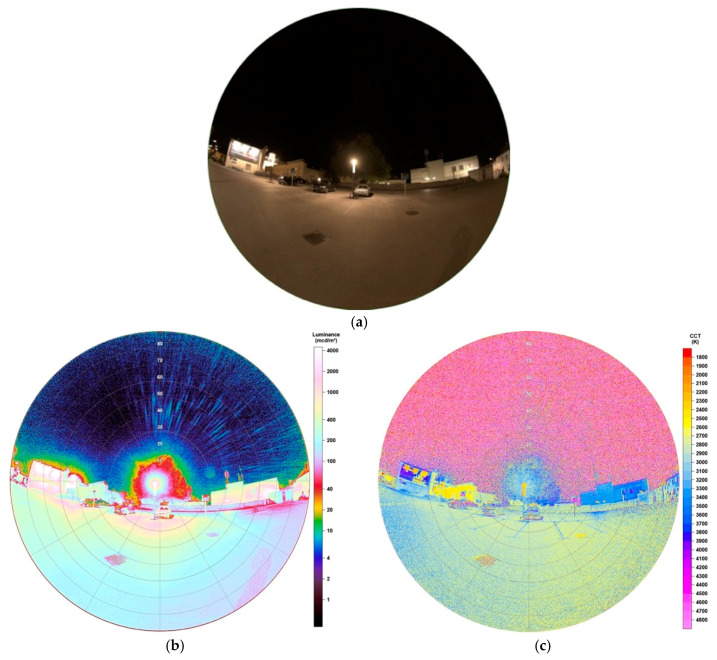
(**a**) Image of tube lamp during night-time taken with fisheye lens for detailed analysis (date: 29.07.2019, 23:23 h; exposure time: 0.3 s); (**b**) Luminance matrix of image (a) as displayed by SQC; (**c**) CCT matrix of image (a) as displayed by SQC.

**Figure 9 jimaging-05-00086-f009:**
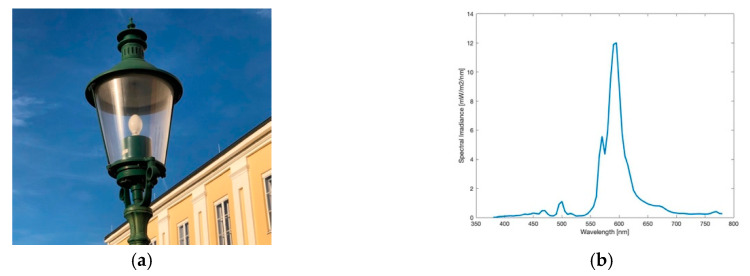
(**a**) Image of the lantern during daytime; (**b**) Spectrum of lamp observed.

**Figure 10 jimaging-05-00086-f010:**
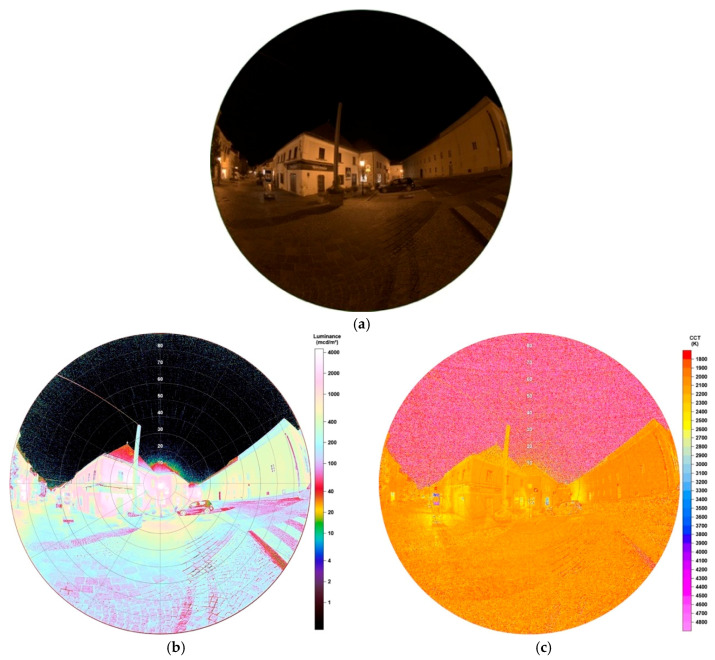
(**a**) Image of lantern during night-time taken with fisheye lens for detailed analysis (date: 29.07.2019, 23:29 h; exposure time: 0.1 s); (**b**) Luminance matrix of image (a) as displayed by SQC; (**c**) CCT matrix of image (a) as displayed by SQC.

**Figure 11 jimaging-05-00086-f011:**
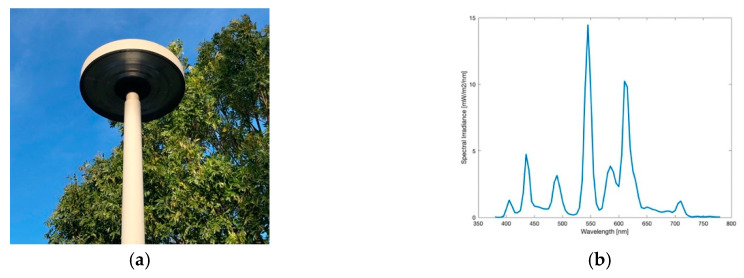
(**a**) Image of the ring-shaped light during daytime; (**b**) Spectrum of lamp observed.

**Figure 12 jimaging-05-00086-f012:**
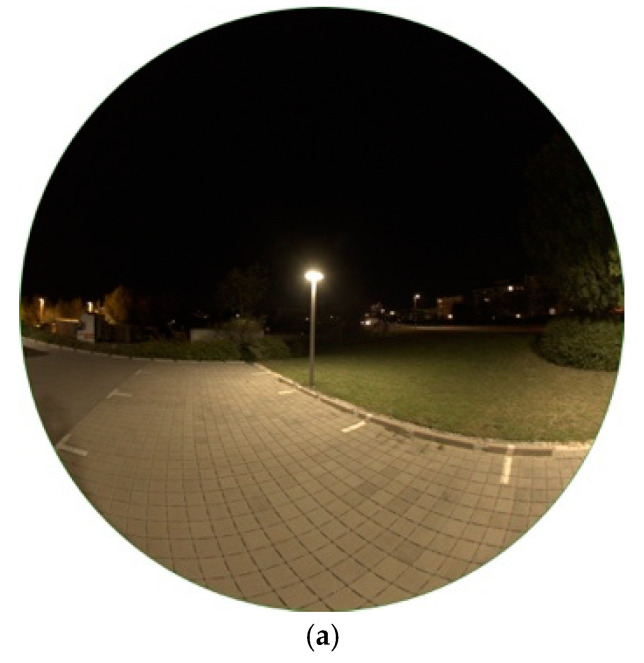
(**a**) Image of ring-shaped lamp during night-time taken with fisheye lens for detailed analysis (date: 24.10.2019, 20:52 h; exposure time: 0.3 s); (**b**) Luminance matrix of image (a) as displayed by SQC; (**c**) CCT matrix of image (a) as displayed by SQC.

**Figure 13 jimaging-05-00086-f013:**
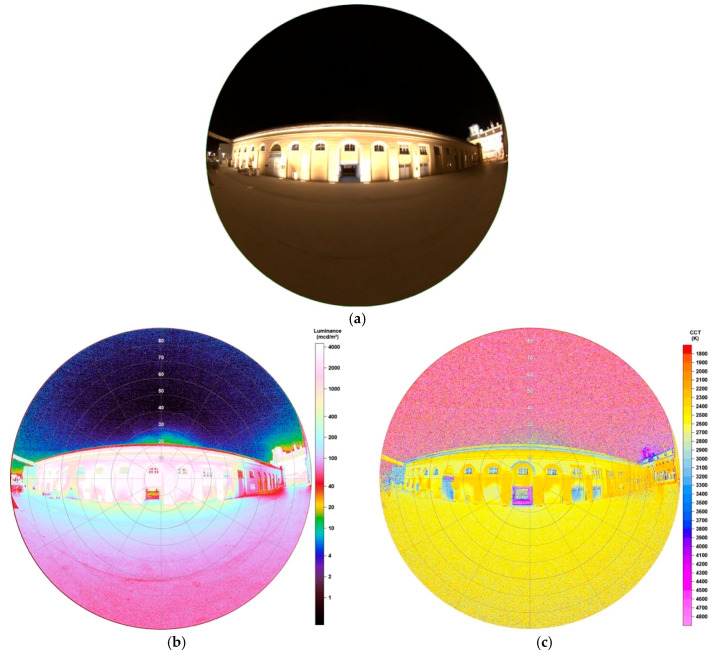
(**a**) Image of the illuminated Market Hall during night-time, taken with fisheye lens for detailed analysis (date: 29.07.2019, 23:24 h; exposure time: 0.4 s); (**b**) Luminance matrix as displayed by SQC; (**c**) CCT matrix as displayed by SQC.

**Figure 14 jimaging-05-00086-f014:**
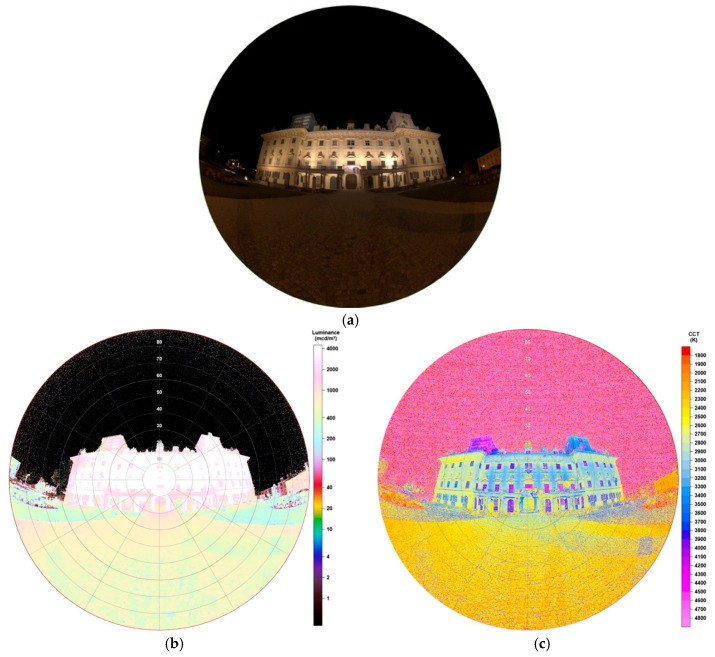
(**a**) Image of the illuminated Castle Esterházy during night-time, taken with fisheye lens for detailed analysis (date: 29.07.2019, 23:25 h; exposure time: 1/30 s); (**b**) Luminance matrix as displayed by SQC; (**c**) CCT matrix as displayed by SQC.

**Figure 15 jimaging-05-00086-f015:**
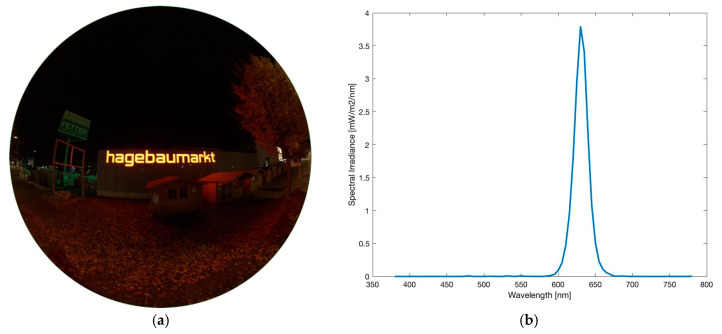
(**a**) Image of the illuminated monochromatic billboard during night-time, taken with fisheye lens for detailed analysis (date: 24.10.2019, 21:20 h; exposure time: 0.1 s); (**b**) Spectrum of billboard observed; (**c**) Luminance matrix as displayed by SQC; (**d**) CCT matrix as displayed by SQC. Please note a higher CCT scale up to 7800 K.

**Figure 16 jimaging-05-00086-f016:**
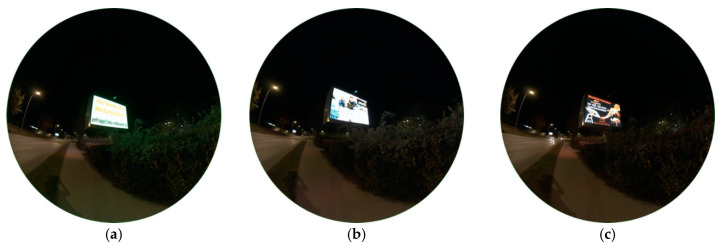
Raw versions and SQC products of images of a LED video wall, all taken on 24.10.2019, 21:09 h with different advertisements: (**a**–**c**) show the raw images captured, (**d**) and (**g**) show luminance and CCT values of advertisement 1, (**e**) and (**h**) of advertisement 2, (**f**) and (**i**) of advertisement 3. Please note a higher CCT scale up to 7800 K.

**Figure 17 jimaging-05-00086-f017:**
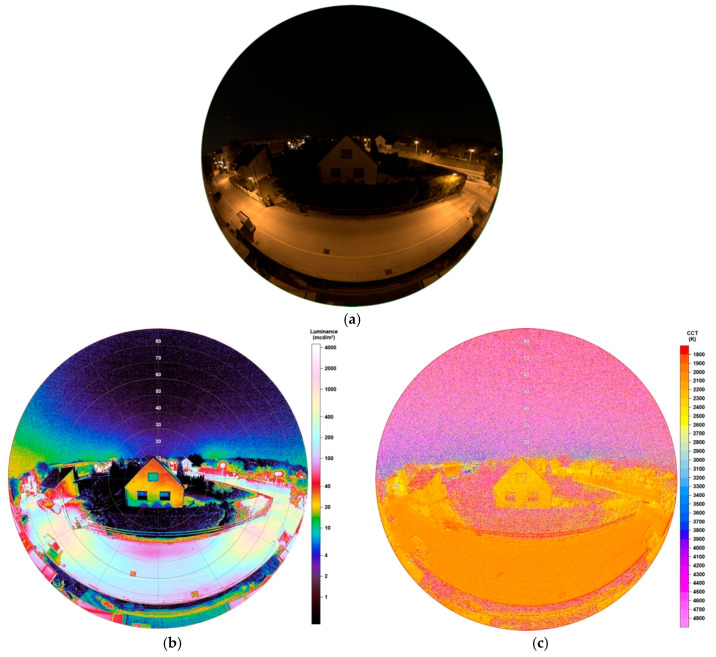
(**a**) Image of illuminated street and building during night-time before conversion to full cut-off-LEDs, taken with fisheye lens for detailed analysis (date: 30.07.2018, 23:22 h; exposure time: 0.5 s); (**b**) Luminance matrix as displayed by SQC; (**c**) CCT matrix as displayed by SQC.

**Figure 18 jimaging-05-00086-f018:**
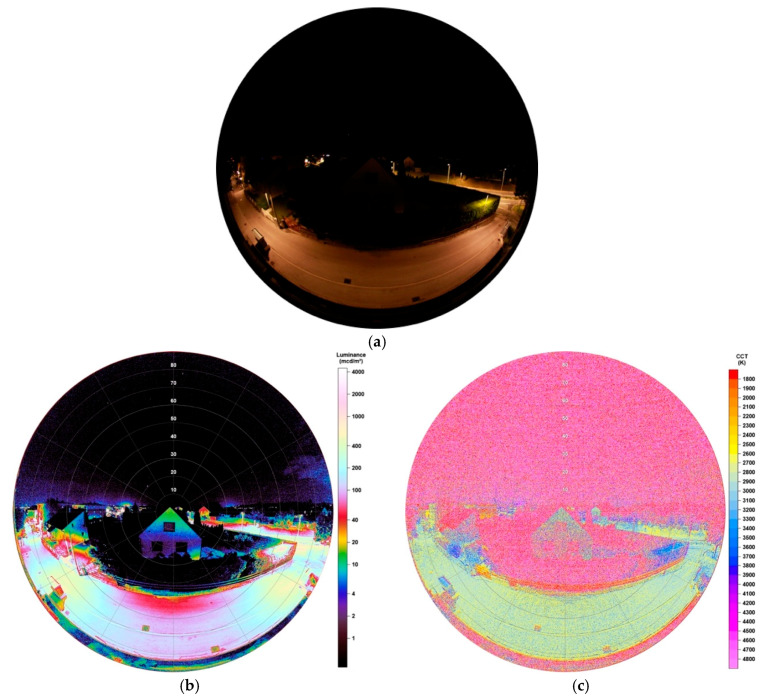
(**a**) Image of illuminated street and building during night-time after conversion to full cut-off-LEDs, taken with fisheye lens for detailed analysis (date: 02.07.2019, 02:03 h; exposure time: 0.5 s); (**b**) Luminance matrix as displayed by SQC; (**c**) CCT matrix as displayed by SQC.

**Figure 19 jimaging-05-00086-f019:**
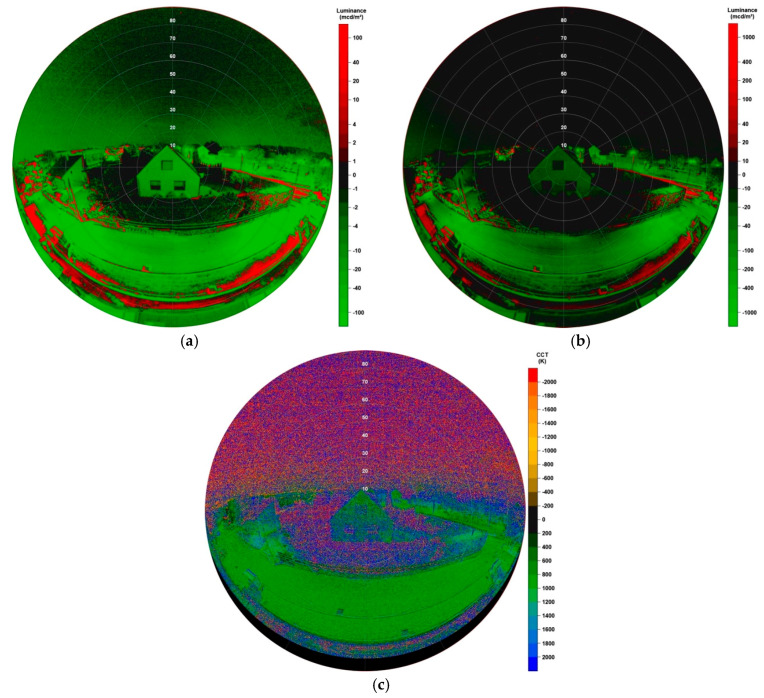
Subtraction of image after transition and image before transition, showing the differences in luminance and CCT. (**a**) Shows luminance comparison with lower scale; (**b**) shows luminance comparison with higher scale; (**c**) shows CCT comparison. Calculation was image after transition (Figure 18) minus image before transition (Figure 17).

**Table 1 jimaging-05-00086-t001:** Development of luminance [mcd/m^2^] and colour temperature [K] values for different zenith angles as analysed from the subtraction of images as seen in Figure 19 by SQC software.

Angle from Image Centre	∆mcd/m^2^	∆CCT (K)
0°–30°	−10.97	+767
0°–60°	−17.89	+759
0°–90°	−28.81	+779
30°–60°	−20.43	+765
59°–61°	−56.77	+835

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
