# Peer review of "Usage of Vertical Fisheye-Images to Quantify Urban Light Pollution on Small Scales and the Impact of LED Conversion"

_2313-433X, 2019, doi:10.3390/jimaging5110086_

Round 1

Reviewer 1 Report

This paper is significantly improved since it was first submitted. t’s a useful proof of concept for an imaging-based measurement/evaluation protocol, especially with the added observations of particular lamp types. Now as a result it is more appropriate for this particular journal. As a result, I recommend that this paper be accepted with a few further changes.

The paper no longer makes sweeping conclusions about the “success” of the LED conversion — this is good. But it does make clear in a limited sense (the one example in section 4) that certain aspects of the conversion were successful, in that now there are lower illuminances and better control of light spill, although at some expense (the higher CCT of the sources). That is also a useful result.

The procedure for computing CCT from images is still not described, even to the point of referring to source information. I remain concerned about this in the sense that the reliability of the CCT measurements presented in the paper hangs on this point. And there are some problems with that I point out in detailed line-by-line comments below. 

65-66    The construction “harvests a total of 14,000 inhabitants” in English is awkward. I would suggest something like “is home to a total of 14,000 inhabitants” or similar.   70    The meaning of “as propagated by the municipal authorities” is unclear in this sentence. Do you mean that the characteristics of the LEDs used were selected or chosen by the authorities?   71    Uplight Output Ratio (ULOR) needs a definition here for the benefit of non-specialist readers.   76    The use of the word ‘optimal’ here is not quite clear to me. Do you mean that Eisenstadt is "optimal” in the sense that there are many light sources there that were NOT subject to the LED conversion? (i.e., that they serve as important points of reference)   93    By “blend of 3.5”, do you mean that the f/ratio of the camera was fixed for all exposures? The meaning of “blend” here is not quite clear.   142    Here and in a few other places in the text the term “light efficiency” is used. It would be better to use instead “luminous efficacy”, which is more widely employed in the lighting industry and beyond.   143    “the typical peak of sodium emissions at 819nm” does not seem right — do you mean the sodium “D line” near 589 nm? The spectrum plotted in Figure 1(b) cuts off well before 819 nm.   148    In the figures beginning with 1(b), plots of the spectral power distribution of the sources in the corresponding images are shown, but no mention is made of how these measurements were obtained. Are SPDs part of the output of Sky Quality Camera? Otherwise, where did these SPDs come from? Also, in Figure 1(c), is the arc-shaped light in the lower right corner of the image a lens flare resulting from the light source near center?   153    Since the procedure by which Sky Quality Camera determines CCTs from images is still not discussed in this version, can you at least comment on how reliable the CCT measurements are as a function of the signal-to-noise ratio of the source imagery? In this example, the sky is about 4600 K. But the S/N of the sky pixels must be quite low. How believable is that CCT, then?   169    You attribute the difference in CCT between the light source viewed directly and surfaces illuminated by the source to saturation of the detector pixels that define the source. At what point does the transition take place for your camera? i.e., at what values of the luminance do you think the pixels begin to saturate? How does that affect both the resulting luminances as well as other parameters like CCT? And to what upper limit of luminance do the CCT values remain reliable?   183    A claim is here stated that “increasing the light efficiency also increases the color temperature”, but this is not as correct now as it was in the past, when the only way to reach high luminous efficacies was by converting less light from the underlying blue LEDs to white light through fluorescence from a phosphor mix. CCT is now less correlated with luminous efficacy, and, e.g., many 3000 K white LED products achieve luminous efficacies within 1-2% of 4000 K and higher.    189    In Figure 4(b), it seems that the CCT of the surfaces illuminated by the ~3200 K source is only about 2700 K. Is that the result of a wavelength-dependent surface albedo? It is certainly true for the grass because of the absorption spectrum of chlorophyll. Compare this view with Figure 2(b), where there seems to be little CCT variation across surfaces of more than one type.   206    It is unclear what “mixing in color temperatures … between the light and illuminated surfaces” means in this context. It sounds like a shortcoming of the software in determining CCT. I would like to see this explained more clearly.   220    With regard to the “HPS-like” SPD in Future 5(b), is the high-pressure sodium spectrum here just simply modified by transmission through the globe cover over the light, which certainly is absorbing some of the light in a wavelength-dependent way?   335    “Full cutoff LED are predestined to emit light only to an apex angle of 70º”. This is not quite right under the (now-deprecated) IES definition of “full cutoff”. Before IES TM-15-07, “full cutoff” was defined as no light emitted above 90 degrees from nadir, and less than 10% emitted at angles above 80 degrees from nadir. So the way this sentence is worded here is not quite right.   350-360    It may be worth mentioning briefly here whether or not the city has any local laws regulating the brightness of building façades, or at least how these numbers for measured façade luminance compare to similar situations, whether in Austria or elsewhere in Europe.   450    In Figure 16, it would be helpful to add a third row showing the raw versions of the images used to make the luminance- and CCT-calibrated versions in (a-c) and (d-f), respectively. This will make the figure qualitatively more like the presentation in all the other figures in the paper.   524-252    “…the night sky seems to have darkened.” But is that simply because of an absence of moonlight in the second image, and not having anything to do with lighting on the ground?   550-558    The discussion here seems to imply higher CCT as the result, but at a lower illuminance, and this is qualified as a success. Was that the expected outcome?   563-564    I don’t understand this table or its necessity to the text. Why are angular distances in the image away from center important?   

Reviewer 2 Report

First of all I have to say that is an improvement from original version and give the chance to use a well known technique on a different way. The examples and tests are now more complete and can give a better idea of the capabilities of this fisheye technique used for lighting purposes.

I have some comments and suggestions to improve the current version. Specially I missed the conclusions section, now merged with discussions. So I recommend some reorganization of the text, maybe some discussion made on results can go to discussion and conclusions moved to their own conclusions section. I will go step by step with my comments.

* Introduction

The introduction is perfectly focused and give a global and accurate perspective for the reader before going inside methodology. Maybe I only missed a reference to manufacturer of SQM so reader can go to know more details of the instrument, it could be just the website.

* Materials and methods

This section has improved clearly specially on the subsection 2.2 where the technical questions are quite clear and also the particular use is described. But all the other subsections are also very well written now.

* Results of various lighting systems

This is clearly the major improving, now it is a good combination of possible uses of this technique with pictures, graphs and explanations of all the cases. So the improvement is excellent. Just some comments.

- In general some of the description is more close to discussion, so maybe part of the results text or a summary of them can go to discussion or conclusions that now are extremely short.

- In all the cases the image (raw image) of lamp at nighttime is too much small. Now appears in combination with lamp image and spectra in a triple panel figure. I think it is interesting to see in a better size the raw images that will be used in the next figures as base of the data. This happens in figures 1, 3, 5, 7, 9 and 11.

- In subsection 3.2 there are the (b) panels of figures 13 and 14 that needs a rescale of the luminance to appreciate better what is explained.

- In figure 15, because it is a sign ‘monochromatic’ it could be very good to add an spectra and it is done with single lamps to compare what is showing with the SQC results.

- In figure 16 the video wall showed three advertisings. Author explain that appear different results depending of the advertising but we have now information about these advertisings. Maybe raw image of each one could be an option of it is not a good idea to not generation promotion of any company a short description of each advertising can be enough (dominant color, etc).

* Results on replacement of lamps

Now this section is clear and focused with the idea of the comparison of lamps before/after replacement. So it is clearly improved. As I said before maybe this could be more on the discussion section because author discusses the change of illumination and how it is seen with fisheye imagery. In addition to this general comment I would propose to analyze with more detail the part of images where there is major changes. SQC software allows this local studies easily.

* Discussion

As I said this section is more or less the conclusions, so some reorganization of section could be an improvement, but the content is ok. Just one comment, when author refers to possible studies with weather conditions as fog or clouds it could be good to refer existing works on fisheye imagery as the one already cited of Jechow et al (reference 25).

* Conclusions

There is no conclusions sections. I propose to move what is really conclusions from discussions to here.

Round 2

Reviewer 2 Report

Dear author,

With this updated version the paper is more consistent and clear, so I will not propose any extra comments or suggestions.